# Assessing Cardiac Functions of Zebrafish from Echocardiography Using Deep Learning

Mao-Hsiang Huang [1] , Amir Mohammad Naderi [1], Ping Zhu [2,3], Xiaolei Xu [2,3] and Hung Cao [1,4,5,*]

1 Department of Electrical Engineering and Computer Science, University of California, Irvine, CA 92697, USA; maohsiah@uci.edu (M.-H.H.); amnaderi@uci.edu (A.M.N.)
2 Department of Biochemistry and Molecular Biology, Mayo Clinic, 200 First St. SW, Rochester, MN 55905, USA; zhu.ping@mayo.edu (P.Z.); xu.xiaolei@mayo.edu (X.X.)
3 Department of Cardiovascular Medicine, Mayo Clinic, 200 First St. SW, Rochester, MN 55905, USA
4 Department of Biomedical Engineering, University of California, Irvine, CA 92697, USA
5 Department of Computer Science, University of California, Irvine, CA 92697, USA
* Correspondence: hungcao@uci.edu

**Abstract:** Zebrafish is a well-established model organism for cardiovascular disease studies in which one of the most popular tasks is to assess cardiac functions from the heart beating echo-videos. However, current techniques are often time-consuming and error-prone, making them unsuitable for large-scale analysis. To address this problem, we designed a method to automatically evaluate the ejection fraction of zebrafish from heart echo-videos using a deep-learning model architecture. Our model achieved a validation Dice coefficient of 0.967 and an IoU score of 0.937 which attest to its high accuracy. Our test findings revealed an error rate ranging from 0.11% to 37.05%, with an average error rate of 9.83%. This method is widely applicable in any laboratory setting and can be combined with binary recordings to optimize the efficacy and consistency of large-scale video analysis. By facilitating the precise quantification and monitoring of cardiac function in zebrafish, our approach outperforms traditional methods, substantially reducing the time and effort required for data analysis. The advantages of our method make it a promising tool for cardiovascular research using zebrafish.

**Keywords:** deep learning; zebrafish; echo-video; ejection fraction; heart disease

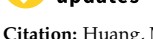



## 1. Introduction

Cardiovascular disease (CVD) constitutes a significant global public health concern. It is a primary contributor to morbidity and mortality, with an estimated 17.9 million fatalities occurring annually, representing approximately 32% of all deaths globally [1]. Notwithstanding substantial financial commitments to medical research, the incidence and prevalence of CVD persist at elevated levels. Therefore, researchers continuously search for novel methods to study heart disease and potential therapies. In recent years, zebrafish (*Danio rerio*) has emerged as a valuable model organism for cardiovascular research [2]. These small fish have numerous advantages over traditional animal models including rapid development, easy maintenance, and genetic tractability. The optically transparent nature of zebrafish embryos also allows in vivo observation of cardiac development, providing researchers with the unique opportunity to scrutinize the intricacies of heart formation in real time [3]. Additionally, the prolific reproduction capabilities of zebrafish facilitate the generation of large numbers of progeny, thus expediting the investigation of genetic variations and their subsequent impact on cardiac function.

Notably, the considerable physiological homology between zebrafish and humans renders this aquatic species a valuable model organism for advancing our understanding of cardiovascular processes. The zebrafish cardiac action potential phenotype is similar to that of humans because they both have a long plateau phase [4]. Moreover, zebrafish can

regenerate damaged heart tissue like certain amphibians, which could provide essential insights into developing therapies for human heart disease [5]. Zebrafish are also an invaluable tool for testing potential therapies for heart disease as researchers can induce heart disease in zebrafish through genetic manipulation or chemical exposure and subsequently examine potential drugs' efficacy in enhancing heart function [6]. For instance, zebrafish have been utilized to evaluate the effectiveness of prospective antiarrhythmic drugs [7].

In the last decade, a panel of cardiomyopathy models has been generated in adult zebrafish, including those modeling inherited cardiomyopathies associated with causative genes such as Myosin heavy chain 7 (MYH7) and LAMP2 [8–10]. However, access to medical imaging data, which is fundamental for constructing such a model, is limited. This is especially true for echo-videos, critical for assessing the heart's function and structure. The manual annotation of echo-videos, which is only proper to be labeled by expert biologists, presents a significant burden, as it is both time-consuming and labor-intensive while also subject to inconsistencies and challenges in validation if labeled by an untrained individual. In light of recent advancements in deep learning methodologies, it has become feasible to devise automated segmentation tools tailored to medical imaging data, encompassing echo-videos. As a subfield of artificial intelligence, deep learning capitalizes on intricate algorithms and neural networks, facilitating continuous learning and enhancement through experiential data. Deep learning algorithms can be meticulously trained in medical imaging to recognize and segment particular structures within images such as the cardiac region in echo-videos [11]. Image segmentation is a crucial task in computer vision and medical imaging that aims to partition an image into different regions based on its semantic meaning. Deep learning-driven image segmentation techniques have demonstrated exceptional efficacy across many applications in recent years. A prominent deep learning architecture employed for image segmentation is the encoder–decoder network, which encompasses two distinct components: an encoder network responsible for extracting high-level features and a decoder network dedicated to producing a pixel-wise segmentation map [12].

The Unet architecture is widely utilized for image segmentation and has become a standard reference for evaluating the performance of image segmentation algorithms [13]. Unet consists of a contracting path for context capture and a symmetric expanding path for precise localization. This unique design enables Unet to capture global context information while preserving detailed spatial information, making it practical for segmenting objects of various sizes and shapes. To enhance the segmentation performance of Unet, Unet++ was proposed as an advanced version [14]. Unet++ utilizes nested and dense skip connections between the encoder and decoder networks, enabling it to capture multi-scale features and achieve better localization accuracy than Unet. In addition, Unet++ employs deep supervision, which involves adding auxiliary segmentation branches to intermediate layers of the network, enhancing training stability and accelerating convergence, leading to better segmentation performance.

Deep learning-based image segmentation architectures, including Unet and Unet++, have demonstrated remarkable promise for medical imaging applications. Furthermore, unsupervised image/video segmentation models have been applied to medical datasets. These models allow for the precise and efficient segmentation of various objects in medical images, thereby aiding in the development of automated and dependable medical diagnosis and treatment systems. In addition, this research has conducted experiments with unsupervised segmentation and supervised image segmentation. Among these, only supervised image segmentation has demonstrated sufficient accuracy for the segmentation task and the evaluation of ejection fraction.

## 2. Materials and Methods

### 2.1. Experimental Animals

Zebrafish (*Danio rerio*) were maintained under a 14 h light/10 h dark cycle at 28.5 °C. All animal study procedures were performed in accordance with the Guide for the Care and Use of Laboratory Animals published by the U.S. National Institutes of Health (NIH

Publication No. 85-23, revised 1996). Animal study protocols were approved by the Mayo Clinic Institutional Animal Care and Use Committee (IACUC #A00002783-17-R20).

*2.2. Dataset*

The present study utilized a dataset consisting of echo-videos of zebrafish. The production of video files was achieved by utilizing the Vevo 3100 high-frequency imaging system, which is equipped with a 50 MHz (MX700) linear array transducer (manufactured by FUJIFILM VisualSonics Inc. Toronto, ON, Canada). This advanced imaging system was employed to precisely measure cardiac function indices in adult zebrafish of varying ages and mutant types. To ensure the highest level of image clarity and detail, we applied acoustic gel (specifically, Aquasonic® 100, produced by Parker Laboratories, Fairfield, NJ, USA) to the surface of the transducer, promoting optimal coupling between the transducer and the tissue interface. To obtain these images, we anesthetized the adult zebrafish using a 0.02% Tricaine (MS 222) solution, which lasted approximately five minutes. Subsequently, each zebrafish was placed ventral side up and held firmly, yet gently, in place using a soft sponge stage.

Regarding image acquisition, the 50 MHz (MX700) transducer was positioned directly above the zebrafish, allowing for the clear capture of images from the sagittal imaging plane of the heart. We secured B-mode images within an imaging field of view of 9.00 mm in the axial direction and 5.73 mm in the lateral direction. Additionally, we maintained a frame rate of 123 Hz, used medium persistence, and set the transmit focus at the heart's center to ensure the utmost image clarity and accuracy. Our data acquisition and subsequent processing followed protocols as outlined in a report available in the existing literature [15]. The application of doxorubicin was also involved, a compound recognized for its capacity to induce cardiomyopathy in adult zebrafish [16,17]. We administered doxorubicin intraperitoneally at a dosage of 20 mg/kg. The ejection fraction decline, a key measure of cardiac function, became detectable via echocardiography 56 days post-injection (dpi). In the orientation of the zebrafish within the echocardiography apparatus, we adhered to a consistent protocol whereby the zebrafish was oriented with its head to the left and ventral side facing upwards. This consistent positioning facilitated reproducibility and standardization across all imaging procedures.

A total of 1005 frames were selected from 164 videos, with a range of 2 to 10 frames extracted from each video (Figure 1a). These frames underwent manual annotation using Vevo Lab software by our team member who is a well-trained biologist working in the field. The cyan lines in the image depicted ventricular boundaries, the long axis (LAX), and the short axis (SAX) of the ventricle (Figure 1b). Following the annotations provided, masks representing the ventricular area were produced through ImageJ software for training the image segmentation model (Figure 1c). We employed data augmentation methods, resulting in a four-fold increase in the dataset's size. We utilized group k-fold cross-validation (k = 5) in data splitting and our model evaluation. Importantly, it was ensured that the validation set contained frames from videos distinct from those in the training set. This step aimed to avoid potential overfitting, wherein the model achieves high accuracy only by extracting features from familiar patterns and signal noise.

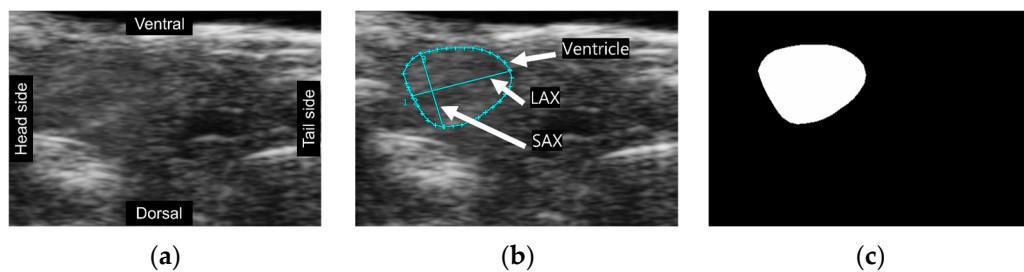

**Figure 1.** (**a**) A cropped frame extracted from echo-videos; (**b**) A cropped image labeled by VEVOLAB; (**c**) A mask created by ImageJ and image processing method.

### 2.3. Cardiac Function Assessment

Ejection fraction (*EF*), an essential metric to evaluate heart function, is quantified as the ratio of blood ejected from the ventricle with each heartbeat and can be mathematically expressed as follows:

$$EF\% = \frac{(EDV - ESV)}{EDV} \times 100\%. \tag{1}$$

The end-diastolic volume (*EDV*) and end-systolic volume (*ESV*) represent the ventricular volumes at the end of diastole and end-systole, respectively. The area–length method is frequently utilized to calculate *EDV* and *ESV* [18] using the following formula:

$$EDV = \frac{8 \times Area(diastolic)^2}{3\pi \times LAX(diastolic)}, \tag{2}$$

$$ESV = \frac{8 \times Area(systolic)^2}{3\pi \times LAX(systolic)}. \tag{3}$$

This method involves the measurement of the area of the ventricle and the length of the ventricular long axis (*LAX*), which is the line connecting the middle of the base of the heart to its tip.

### 2.4. Unsupervised Segmentation Approach

Unsupervised image and video segmentation have shown considerable promise in computer vision tasks. The challenge is identifying which method can be applied for extracting features for zebrafish echo-videos. Modified convolutional neural networks (CNNs) [19] have been used (Figure 2c), which assign labels to pixels without requiring labeled data. The pixel labels and feature representations are optimized through gradient descent to update the network's parameters.

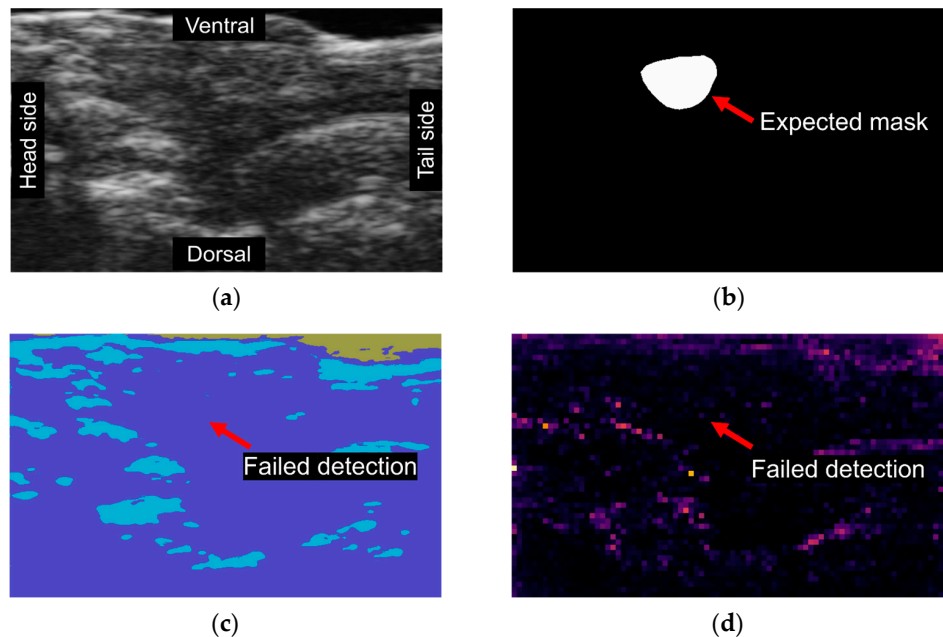

**Figure 2.** Unsupervised segmentation trials utilizing different methods failed to detect expected masks. Each image is represented by a distinct color, signifying the model-generated segmentation regions. (**a**) An original frame, where individuals without training could not accurately classify the ventricle's location due to the lack of ground truth labels; (**b**) An expected mask created manually; (**c**) A result frame obtained using modified CNNs revealing that the model is susceptible to color interference and, as such, is unsuitable for echo-videos; (**d**) A result of Dino demonstrating the model was incapable of detecting the ventricle due to background noise.

Dino, an implementation of self-supervised learning on Vision Transformers (ViTs) [20], was applied for unsupervised video segmentation(Figure 2d). Dino employs a simplified self-supervised training approach by predicting the output of a teacher network. It comprises a momentum encoder and uses a standard cross-entropy loss. Dino has two key features that distinguish it from CNNs and supervised ViTs: it explicitly encodes semantic information about image segmentation such as scene layout and object boundaries.

However, these methods are not robust enough to handle echo-video's blurring and signal noise, making them unsuitable for the image segmentation task and ejection fraction assessment. As a result, supervised image segmentation has been the only method to demonstrate sufficient accuracy in this work.

### 2.5. Supervised Image Segmentation Approach

In our previous work, we proposed the ZACAF, a framework based on a deep learning model for automated assessment from bright field microscopy videos [21]. This study employed various deep learning models to perform image segmentation on a given dataset, utilizing segmentation model modules on the PyTorch platform. Resnet [22], Efficientnet [23], ResNeXt [24], and Mobilenet [25] were selected as the encoder components. Resnet34, Efficientnet-b4 (Figure 3), and ResNeXt-50-32x4d were chosen to enable comparative analysis and investigate which architecture could extract better features from this dataset with comparable parameter sizes. Resnet is a widely used architecture with a relatively shallow depth, while Efficientnet has demonstrated excellent performance in various image classification tasks due to its hierarchical structure. ResNeXt, on the other hand, employs a split-transform-merge strategy to aggregate information from multiple pathways. In addition, Mobilenet-v2 was selected for comparison due to its lightweight design, which makes it a practical choice for mobile and embedded devices.

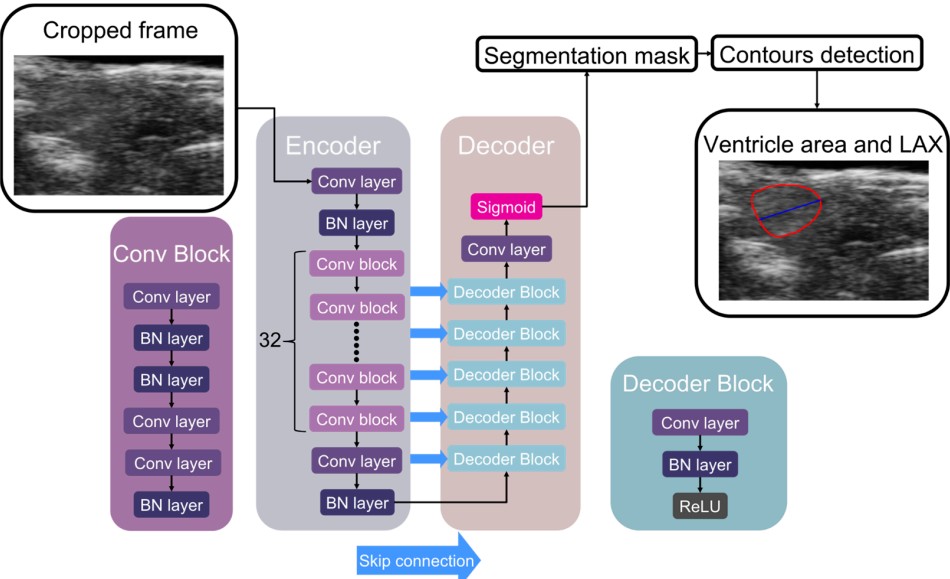

**Figure 3.** The selected example supervised image segmentation model architecture and its associated process flow. Specifically, we utilized Efficientnet-b4 as the Encoder and Unet as the Decoder in this figure. Initially, we cropped the frame from echocardiography videos and fed it into the Encoder. The Encoder comprised a convolution layer (Conv layer), batch normalization layer (BN layer), and Convolution block (Conv Block), each consisting of several convolution layers and batch normalization layers with varying scales. Next, we utilized the Decoder, which included the Decoder block, convolution layer, and sigmoid function. Each decoder block contains a convolution layer, batch normalization layer, and rectified linear unit (ReLU) activation function. Upon completing the deep learning model, we obtained a segmentation mask. Subsequently, we utilized contour detection to locate the ventricle, with the LAX determined via the midpoint of the left base connected to the right tip of the heart.

Unet (Figure 3) and Unet++ were chosen as decoders for the image segmentation decoder component. The training process was executed using NVidia's A10 and H100 GPU, which was provided by an online cloud service. The dice loss function, a widely used semantic image segmentation loss function, was applied to train the models [26]. This loss function is particularly suitable for medical image segmentation tasks, as it addresses the imbalance between foreground and background classes. It penalizes false negative and false positive errors equally, guaranteeing precise segmentation results. In contrast, we discovered that using pre-trained weights, often employed to enhance the performance of deep learning models by transferring knowledge from large-scale datasets, negatively impacted the model's accuracy in several experiments. This could be attributed to the domain shift between the pre-trained and specific medical datasets used in this work.

*2.6. Quantitative Comparison of Approaches*

2.6.1. Dice Coefficient

The dice coefficient (DC) is a commonly used evaluation metric in image segmentation tasks. It measures the degree of similarity between two objects, where a score of 1 denotes perfect agreement or complete overlap, and a score of 0 indicates no overlap. The calculation of the dice coefficient is obtained by taking twice the intersection of the two objects and dividing it by the sum of the pixels in both objects. In the case of binary segmentation, the formula can be expressed as:

$$Dice = \frac{2|A \cap B|}{|A| + |B|}, \tag{4}$$

where *A* and *B* represent the two objects being compared and the absolute values of *A* and *B* denote the total number of pixels in each object. The intersection of the two objects is the number of pixels that both objects have in common.

2.6.2. Intersection over Union

Intersection over union (IoU), also known as the Jaccard Index, is a widely used metric for evaluating the performance of image segmentation algorithms. This metric is calculated by taking the ratio of the area of overlap between the predicted segmentation and the ground truth segmentation to the area of union between them. IoU ranges from 0 to 1, with 0 indicating no overlap and 1 indicating perfect overlap. For binary segmentation, the IoU can be computed using the following formula:

$$J = \frac{|A \cap B|}{|A \cup B|}, \tag{5}$$

where *A* and *B* represent the predicted and ground truth segmentation masks, respectively. The intersection between *A* and *B* refers to the set of pixels where both *A* and *B* have a non-zero value, while the union between *A* and *B* refers to the set of pixels where either *A* or *B* has a non-zero value. This metric is commonly used in deep learning-based segmentation models, as it provides a reliable measure of the accuracy of the model's predictions compared to the ground truth segmentation.

2.6.3. Receiver Operating Characteristic (ROC)

The receiver operating characteristic (ROC) curve serves as a graphic representation of a binary classifier's diagnostic competence as the discrimination threshold is adjusted. For the incorporation of ROC curves in the image segmentation model, our preliminary step was the calculation of a probability map corresponding to the target segmentation. Herein, each pixel is attributed with a probability value indicating its likelihood of being a part of the foreground. As the next step, we adjusted the threshold used to classify a pixel as foreground or background, facilitating the computation of the true positive rate and the false positive rate at each respective threshold.

The area under the ROC curve (AUC), a critical metric, provides a quantitative measure of the model's precision in pixel classification, irrespective of the selected threshold. In an ideal scenario, a flawless classification model would be characterized by an AUC of 1, whereas a model whose performance equates to that of random classification would exhibit an AUC of 0.5. The ROC curve and the AUC together serve as indicators of our image segmentation models.

## 3. Results

The findings depicted in Figure 4 illustrate the model's performance trained using the dice loss function alongside an Adam optimizer and a learning rate of 0.001, incorporating a decay rate of 0.8 every 20 epochs. We employed an early stopping mechanism during model training to cease training when the model's performance on the validation set does not show substantial improvement, thus avoiding unnecessary computational expenditure and the potential for overtraining. The hyperparameters were carefully adjusted to attain the best validation IoU score. The Unet++ and Efficientnet architectures outperformed the other encoder–decoder architectures, achieving validation Dice coefficients of 0.967 and validation IoU scores of 0.937. Efficientnet proved its capability in extracting reliable features from the specific frames despite its comparable parameter sizes to the other encoder architectures. Furthermore, Mobilenet_v2, with its relatively low number of parameters attained a similar IoU score with other encoders, suggesting its effectiveness as a feasible alternative for lightweight segmentation tasks. During the k-fold cross-validation training process, the model demonstrated consistently high performance, with AUC values exceeding 0.93 in all instances. The detailed results of the ROC analysis, along with these AUC scores, were presented in Figure 5.

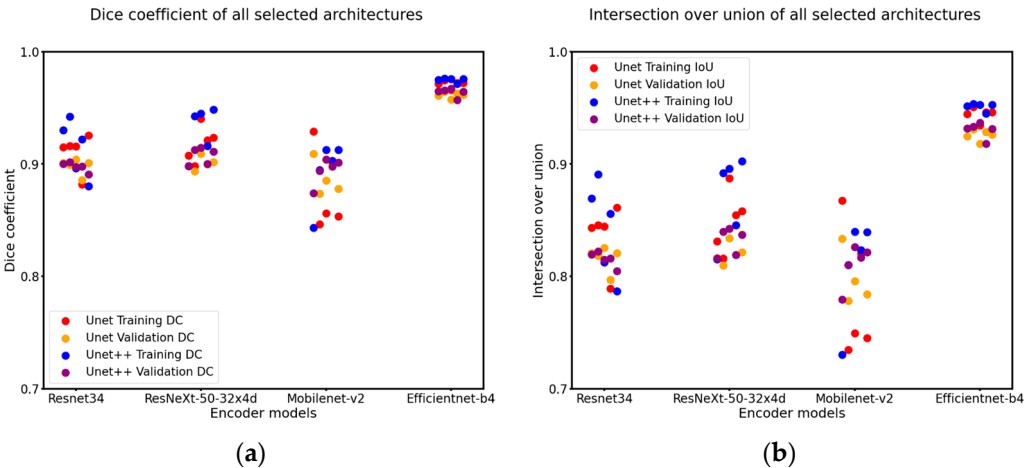

**Figure 4.** The dice coefficient (**a**) and intersection over union (**b**) results with k-fold cross-validation for all selected architectures, with the *x*-axis representing different encoder architectures and the *y*-axis denoting the metrics. Our model selection process was based on the higher validation result achieved by each architecture. The Unet++ and Efficientnet architectures outperformed the rest, achieving a validation Dice coefficient of 0.967 and validation IoU score of 0.937. Notably, overfitting was observed, as indicated by the difference in training and validation metrics across all architectures.

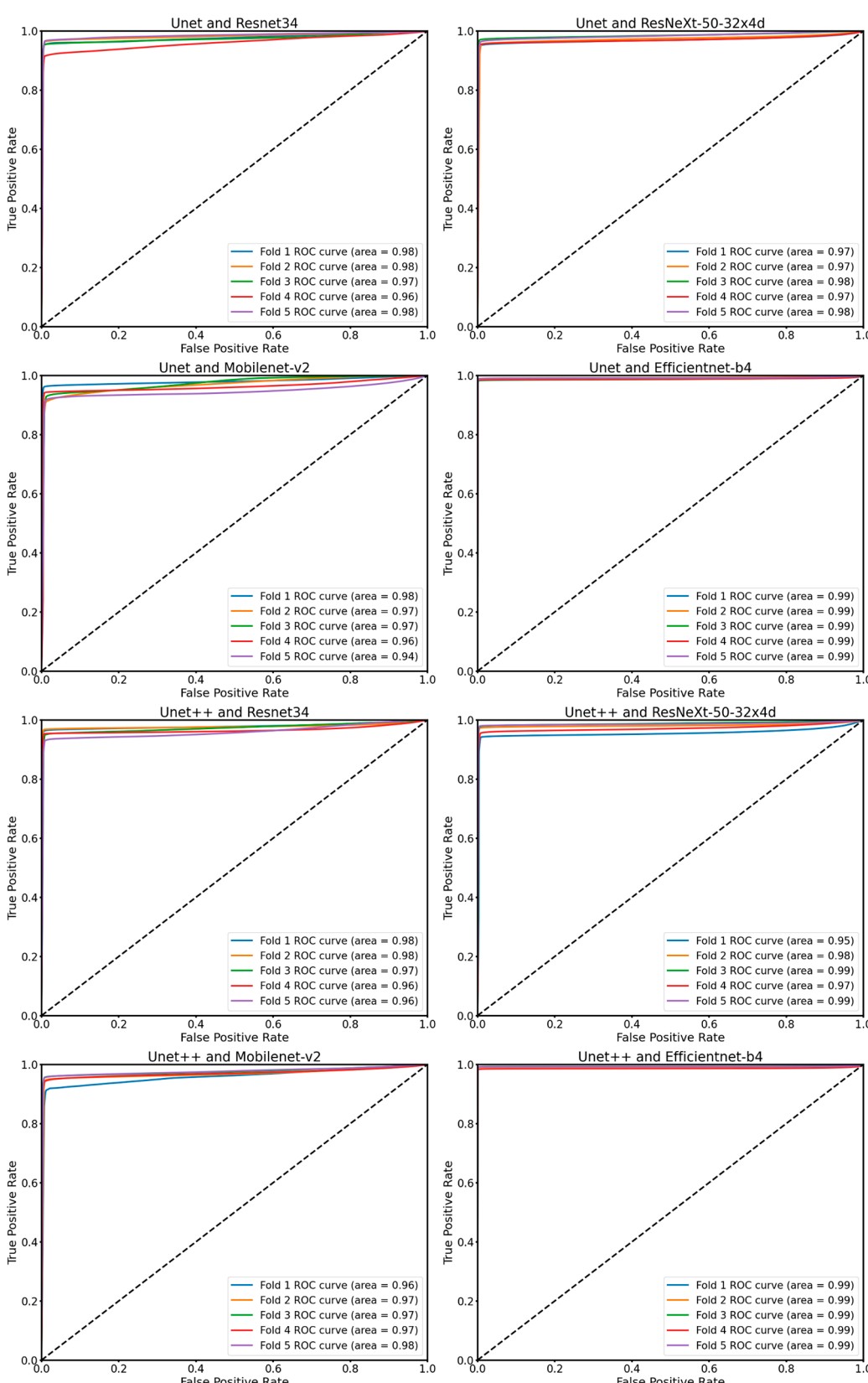

**Figure 5.** The receiver operating characteristic (ROC) curve and the area under the ROC curve during the k-fold cross-validation training process. AUC values exceed 0.93 in all instances.

The present study aimed to develop a method for automatically calculating ejection fraction (EF) in fish populations using image processing techniques (Figure 6). The method's

performance was evaluated by comparing its results with manual assessments of the EF by three experienced biologists. The methodology involved detecting the ventricle area and subsequently measuring its size and the left axis length (LAX) from selected frames. A set of 51 videos containing three groups of fish was utilized for the evaluation, with manually labeled frames being chosen for the analysis. The obtained results indicated that the error rate ranged from 0.11% to 37.05% (Figure 7). Results from our automatic EF evaluation are consistent with manual measurements by each of the three biologists. A significant reduction of cardiac function in the AIC model can be detected (Figure 8). Notably, the average error was 9.83%, indicating the high accuracy of the automatic EF evaluation. The study reveals that the proposed method is promising to enhance the efficiency and accuracy of EF calculation in zebrafish.

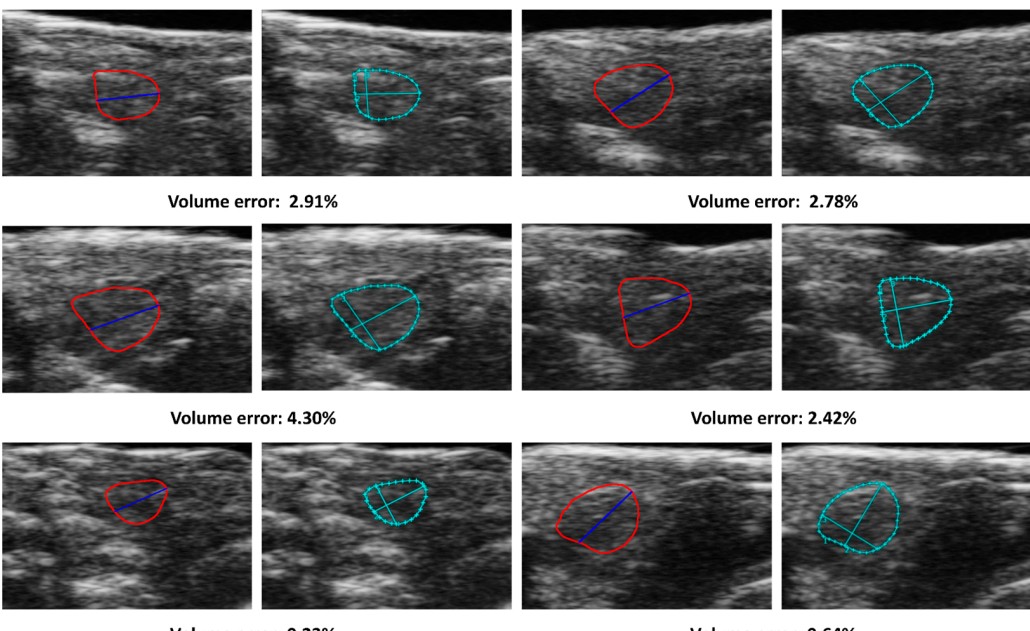

**Figure 6.** Comparison of automatic and manual volumes measurements. The solid lines in the figure corresponded to the ventricle contour and the LAX and detected by the proposed method. The cropped images, labeled manually with dotted lines, correspond to the manual label for each frame. The figures in the same row were selected from the same fish group, and the measurements are in pixels.

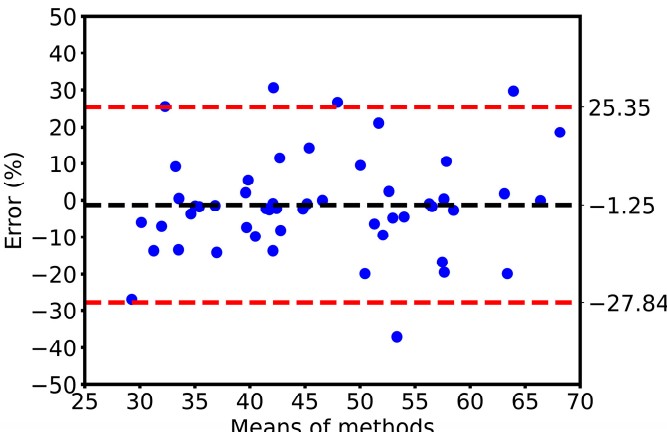

**Figure 7.** Bland–Altman plot for 51 sets of measurements of the EF evaluation using manual and automatic methods. Each point in the figure represents the assessment result from a different video and the measurements are in pixels.

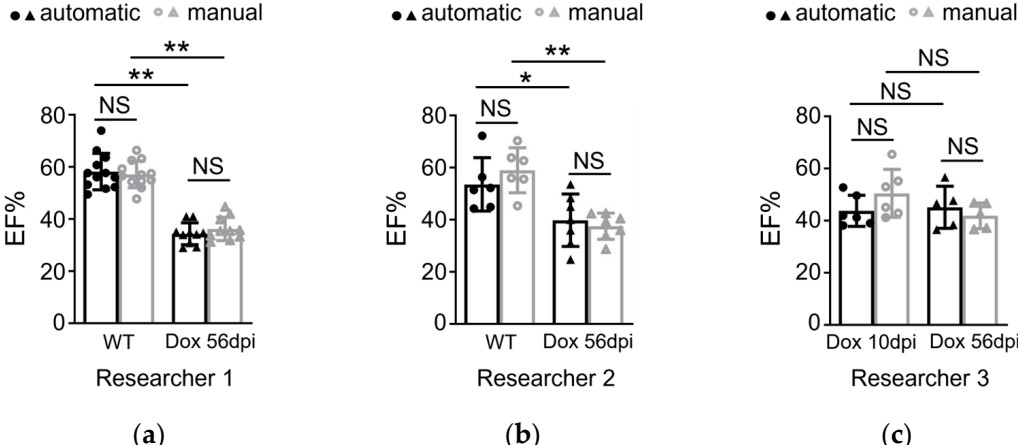

**Figure 8.** (**a**–**c**) Quantification of the ejection fraction in adult zebrafish showed no significant difference by using automatic and manual methods in two different groups by three researchers (*n* = 5–12 per group). Dox group: intraperitoneal injection of DOX dosed at 20 mg/kg, echoed at 10 dpi or 56 dpi. Automatic and manual methods both detected Dox induced cardiac function deduction. Of note, the automatic method was also able to detect reduced EF in 10 dpi Dox fish injected by researcher 3. Data are presented as mean ± SD, * $p < 0.05$, ** $p < 0.01$. NS, non-significant. WT, wild type.

## 4. Discussion

Medical image segmentation is critical in analyzing cardiac function in research and clinical settings. Manual segmentation is time-consuming and prone to variability, making automated segmentation methods highly desirable. In this study, we investigate the segmentation of zebrafish heart imaging to evaluate the efficacy of several segmentation model architectures. Unsupervised learning methods have emerged as promising for image and video segmentation because they can extract features without requiring labeled data. However, the effectiveness of unsupervised learning heavily relies on the quality and complexity of the dataset [27]. The quality of the zebrafish dataset has several limitations that affect the performance of unsupervised learning methods. Firstly, the dataset's small size and blurred nature pose challenges for unsupervised learning, leading to suboptimal segmentation outcomes. Secondly, an untrained individual would have difficulty classifying the dataset due to the variability in the videos' visual appearance. Furthermore, videos from the same fish group can look different even if they are in the same group, making classification challenging.

Combining Efficientnet and Unet++ yielded the highest segmentation results in the zebrafish dataset. Despite implementing data augmentation, overfitting, a common pitfall in machine learning, was observed in the remaining architectures. This technique, commonly employed to enhance dataset size, failed to elevate validation outcomes in our study. The highly specialized dataset, temporal information, and limited dataset variability rendered the data augmentation ineffective in enhancing model consistency. Interestingly, a marked increase in model accuracy was only observed when the encoder was paired with Efficientnet. As for the decoder architecture, Unet generated comparable results to Unet++ despite a notable difference in training time. Unet++ necessitated more than one-third of the total time for training compared to Unet, making the latter a more time-efficient option, especially for newer and smaller datasets. These findings suggest that over-complex architectures may not generalize well to smaller datasets such as the zebrafish dataset.

Lower Intersection over Union (IoU) scores could engender inaccuracies in the ventricle area and LAX measurements. Such inaccuracies precipitate anomalies in ejection fraction (EF) evaluations, given that the demarcated area may not accurately emulate the configuration of the ventricle. The primary causative factor leading to

an error rate as substantial as 37.05% is attributed to the oversight of the ventricle shape during the mask segmentation phase, resulting in a volume calculation that is excessively large in relation to the LAX. Moreover, the Dice loss function, premised solely on the intersection of the two entities, may neglect the inclusion of the shape pattern. This issue underscores the significance of shape preservation during cardiac image analysis. Although unsupervised learning has demonstrated potential in more expansive datasets, such as DAVIS [28], constraints related to the zebrafish dataset's size and quality impede optimal performance. In light of these restrictions, forthcoming research endeavors should examine more comprehensive and superior-quality datasets to further augment the efficacy of unsupervised learning approaches. Furthermore, combining supervised and unsupervised learning methods could improve medical image segmentation accuracy [29]. A supervised segmentation model based on the video model can also improve segmentation accuracy. However, this would require significant amounts of data and extensive labeling efforts, which are not feasible for this study's original proposal. Therefore, future work could also focus on selecting the unsupervised algorithm for extracting the features that support the model training.

## 5. Conclusions

The utilization of deep learning algorithms for the segmentation and assessment of cardiovascular metrics presents a promising research direction due to its potential to improve accuracy, efficiency, and objectivity in analyzing complex biomedical data. The accurate and automatic evaluation of cardiovascular metrics from echo-videos has the potential to support studies in zebrafish models. The implementation of automation in image processing and analysis has the potential to alleviate the burden on researchers while simultaneously enhancing the precision and replicability of data interpretation. Nevertheless, this investigation emphasizes the inherent difficulties and constraints of employing deep-learning methodologies on compact and domain-specific datasets. The paucity of training data may render deep-learning models vulnerable to overfitting, consequently, leading to diminished generalization capabilities and a decline in overall predictive accuracy. Therefore, exploring and optimizing techniques to improve model performance on small datasets is crucial. Overall, developing an automated system for segmenting zebrafish embryos from echo-videos utilizing supervised deep-learning methods advance biomedical research. Further optimization and improvement of deep learning models will enable accurate and efficient evaluation of cardiovascular metrics from echo-videos, supporting research on cardiovascular development, disease diagnosis, and drug discovery in zebrafish models.

**Author Contributions:** M.-H.H., H.C. and X.X. contributed to the concept of the study. M.-H.H. and A.M.N. performed the data analysis. M.-H.H., A.M.N. and P.Z. performed the data interpretation. All the authors contributed to the drafting of the manuscript. All authors have read and agreed to the published version of the manuscript.

**Funding:** The authors would like to acknowledge the financial support from the NIH SBIR grant #R44OD024874 (H.C.), the NIH HL107304 and HL081753 (X.X.).

**Data Availability Statement:** Data available on request due to privacy/ethical restrictions.

**Conflicts of Interest:** The authors declare no conflict of interest.

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
