# Peer review of "Assessing Cardiac Functions of Zebrafish from Echocardiography Using Deep Learning"

_information, doi:10.3390/info14060341_

Round 1
Reviewer 1 Report
The authors of this manuscript present a comparison of two different machine learning algorithms to analyse zebrafish echocardiograms.
I miss some information about whether additional algorithms are available, and why those choosen are prefered.
As for the calculation of EF a picture at systole and one at diastole is needed, using only two pictures for analysis seems a very low number.
Also, I don't understand why the genes myl7 and lamp that are mentioned in the introduction play a role in this manuscript. There is a variety of other genes that are also associated with cardiac disease.
Author Response
Please see the attachment. Thank you for your time.

Reviewer 2 Report
In its current form I cannot recommend this manuscript for publication. Overall, there are considerable unanswered questions concerning both experimental methodology and the deep learning training. As a result, this inevitably affects the repeatability, the applicability of the work, and ultimately the conclusions reached by the authors.
In regard to the experimental serious questions exist including (but not limited to): (i) creation of the video files; (ii) information on the zebrafish used; (iii) details on the echocardiography apparatus; (iii) the application of doxorubicin; and (iv) orientation of the zebrafish in the echo apparatus as presented in the figures.
In regard to the deep learning paradigm concerns exist concerning (but are not limited to): (i) the relatively low number of training frames/videos; (ii) a lack of cross-validation (including lack of ROC curves; lack of multiple user validation; (iii) potential issues with accuracy on other datasets including different animals and different user qualification; (iv) overfitting (ie: is it learning all hearts or simply these hearts); and (v) if the masks are being applied in the appropriate location based on figures presented.
With these limitations considered, the scientific soundness of this work is called under question. If the above concerns are directly addressed, it may be possible to reconsider this manuscript after considerable revisions.
Overall adequate, perhaps minor proofreading/editing required.
Author Response

(The authors gave the same response as above.)

Round 2
Reviewer 2 Report
In terms of the revised manuscript, while significant work was included with resubmission there still remains some question about the impact and repeatability of the work based on the manuscript presented. In short, serious considerations and revisions should still be made to the present work including:
1) If I am understanding the author’s work correctly, all analysis is based on annotation of a single researcher (Line 130-131). It is significant to note if the single individual who annotated the current data was blinded to condition (not currently described that could be found). If this work is to be considered repeatable additional annotation (ideally blinded) should be performed.
2) In terms of the author’s response to the applicability of other animals is one appreciated (and considerable), the concern meant to be raised was whether the AI was learning only those zebrafish in the dataset and if other new zebrafish (eg; different strain/size/age) were introduced if the paradigm would do as well?
3) In the figures dorsal is marked at the top while ventral is marked at the bottom of all panels. In the text the authors describe placing the zebrafish ventral side up (Lines 112-113 and 125-127). This orientation would place the masks in the correct location as described in the author’s response, however the figures labelled as currently with dorsal at top make interpretation potentially overly difficult, and easily misinterpreted, for a reader and could be fixed easily.
None.
Author Response
Please see the attachment. Thank you very much for your time.

Round 3
Reviewer 2 Report
Overall, my feedback has now been sufficiently answered.